# Improved Reverse Transcription Loop-Mediated Isothermal Amplification (RT-LAMP) for the Rapid and Sensitive Detection of *Yam mosaic virus*

**DOI:** 10.3390/v15071592

**Published:** 2023-07-21

**Authors:** Ruth O. Festus, Susan E. Seal, Ruth Prempeh, Marian D. Quain, Gonçalo Silva

**Affiliations:** 1Natural Resources Institute, University of Greenwich, Central Avenue, Chatham Maritime, Kent ME4 4TB, UK; s.e.seal@greenwich.ac.uk (S.E.S.); g.silva@greenwich.ac.uk (G.S.); 2Council for Scientific and Industrial Research-Crops Research Institute, Fumesua, Kumasi P.O. Box 3785, Ghana; r.prempeh@cropsresearch.org (R.P.); m.quain@cropsresearch.org (M.D.Q.)

**Keywords:** yam, seed systems, *Yam mosaic virus*, virus diagnostics, isothermal amplification, RT-LAMP, West Africa

## Abstract

Yam (*Dioscorea* spp.) productivity is constrained significantly by the lack of a formal seed system. Vegetative propagation, through tuber setts as ‘seed’ yams, encourages the recycling of virus-infected planting materials, contributing to high virus incidence and yield losses. Efforts are ongoing to increase the production of high-quality seed yams in a formal seed system to reduce virus-induced yield losses and enhance the crop’s productivity and food security. Specific and sensitive diagnostic tests are imperative to prevent the multiplication of virus-infected materials contributing to a sustainable seed yam certification system. During routine indexing of yam accessions, discrepancies were observed between the results obtained from the reverse transcription loop-mediated isothermal amplification (RT-LAMP) test and those from reverse transcription polymerase chain reaction (RT-PCR); RT-LAMP failed to detect *Yam mosaic virus* (YMV) in some samples that tested positive by RT-PCR. This prompted the design of a new set of LAMP primers, YMV1-OPT primers. These primers detected as little as 0.1 fg/µL of purified RNA obtained from a YMV-infected plant, a sensitivity equivalent to that obtained with RT-PCR. RT-LAMP using YMV1-OPT primers is recommended for all future virus-indexing of seed yams for YMV, offering a rapid, sensitive, and cost-effective approach.

## 1. Introduction

*Yam mosaic virus* (YMV) is a prevalent virus of yam [1,2,3,4], an important staple food crop in many parts of the world [5]. It belongs to the genus *Potyvirus* and has a single-stranded, positive-sense RNA genome that is approximately 9.6 kb long and encodes a single large polyprotein, which is cleaved into smaller proteins [6,7,8,9]. YMV is widely distributed in tropical and subtropical yam-growing regions, particularly in West Africa, the West Indies, and the Caribbean [3,4,10]. It commonly infects *D. rotundata*, *D. cayenensis-rotundata*, and *D. alata* [3,11]. The virus is transmitted through vegetative propagation of infected yam materials or by aphid vectors in a non-persistent manner, causing various symptoms, including mosaic patterns on leaves, stunted growth, and reduced yields [3]. YMV has been reported to cause about 40% yield loss in yam fields [12,13] and hamper the exchange of valuable germplasm for the crop’s improvement.

Yam plays a vital role in food security, income generation, and nutrition for smallholder farmers, especially in West Africa, which produces over 95% of the world’s total yam production [3,5,14]. An infection with YMV in the field threatens the food security and livelihoods of West Africans. The absence of a formal seed yam certification system and farmers selecting small tubers from their harvest for planting the following season encourage the propagation of infected materials, which has been instrumental to the spread of YMV in yam-growing regions [15,16,17]. 

The use of virus-free planting materials is the most effective method to control the spread of viruses infecting yam [18,19]. Several methods have recently been developed to boost the production of virus-free seed yams, including single-node vine cuttings, tissue culture, hydroponics, and aeroponic systems [16,20,21]. The development of sensitive and cost-effective diagnostics is paramount to guarantee the production of virus-free seed yams for a sustainable formal seed system [22,23]. These diagnostics methods must address virus detection challenges, including false-negative results arising from reduced virus titre associated with clonally propagated crops [8]. Furthermore, the genomic variability of YMV makes the detection of all putative isolates/variants challenging [24].

Methods used for the detection of YMV include enzyme-linked immunosorbent assay (ELISA), reverse transcription polymerase chain reaction (RT-PCR), immunocapture-RT-PCR (IC-RT-PCR), and isothermal assays such as recombinase polymerase amplification (RPA) and reverse transcription loop-mediated isothermal amplification (RT-LAMP) [3,22,23,25,26,27,28]. RPA and RT-LAMP offer similar and greater sensitivities, respectively, compared to RT-PCR, with more benefits, including speed, cost-efficiency, in-field diagnosis, and ease of establishment in resource-challenged laboratories, and are considered advantageous for the routine detection of YMV [22,23,29]. 

Routine indexing of yam plants for YMV in our laboratories identified discrepancies between the RT-LAMP [23] and RT-PCR [25] tests. RT-LAMP gave false-negative results for some samples, which were confirmed positive for YMV by RT-PCR and Sanger sequencing. False-negative results could permit the multiplication of infected plant materials in the seed systems, discrediting the integrity of quality seeds distributed to farmers [19,28]. Further, it could pose severe challenges to plant health by spreading viruses or novel variants to new regions through the exchange of infected germplasms [28]. This prompted the development of a new set of LAMP primers which are described in this article and were found to increase not only the specificity but also the sensitivity of YMV detection compared to existing YMV LAMP primers [23].

## 2. Materials and Method

### 2.1. Plant Material, Total RNA Extraction, and Crude Sample Preparation

Yam (*D. rotundata* and *D. alata*) leaf tissues used in this study were obtained from plants grown in glasshouses at the Natural Resources Institute (NRI), United Kingdom, the Centre for Scientific and Industrial Research-Crops Research Institute (CSIR-CRI) in Ghana, and yam field surveys conducted in Benin, Cameroon, Togo, and Nigeria (Table 1). Total RNAs were extracted from leaf tissues using the Spectrum Plant Total RNA Kit (Sigma-Aldrich, Saint Louis, MO, USA), according to the manufacturer’s recommendations. The concentration and purity of extracted yam RNAs were measured using a NanoDrop 2000 spectrometer (ThermoScientific, Waltham, MA, USA). 

The detection of YMV from crude extracts was carried out using the protocol described by Silva et al. [27]. One leaf disc was immersed in 300 µL of freshly prepared PEG buffer (6% *w*/*v* polyethylene glycol (PEG)−200 in 20 mM NaOH). The tubes were vortexed briefly and incubated for 5 min at room temperature. Crude extracts were used directly as templates in RT-LAMP assays.

### 2.2. The Detection of YMV by RT-PCR and Phylogenetic Analysis

The detection of YMV by RT-PCR was carried out using the primer pair YMV CP 1F and YMV UTR 1R (Table 2), which amplifies a 586 bp region comprising a partial coat protein (CP) gene and the 3’ UTR region of the YMV genome [25]. The RNA quality was confirmed by amplifying the yam actin gene, as described by Silva et al. [22]. RT-PCR assays were set up as 20 µL reactions containing 0.2 µM of each primer (Sigma Aldrich), 0.25 mM of each dNTP (ThermoScientific), 1.25 U DreamTaq DNA Polymerase (ThermoScientific), 2.5 U AMV-reverse transcriptase (Promega, Madison, WI, USA), 1X DreamTaq Green Buffer containing 2 mM MgCl_2_ (ThermoScientific), and 2 µL RNA as template. Thermal cycling conditions were 50 °C for 10 min, followed by 95 °C for 4.5 min, and 35 cycles of 95 °C for 30 s, 55 °C for 30 s, and 72 °C for 30 s, and a final extension of 72 °C for 10 min. RT-PCR products were analysed by electrophoresis on agarose gels [2% (*w*/*v*) agarose in Tris-borate-EDTA (0.5 × TBE) buffer (pH 8.0)] and viewed under UV light using a gel doc system (SynGene, Cambridge, UK). PCR products were purified and Sanger-sequenced by the Source BioScience sequencing service (Cambridge, UK). The nucleotide sequences generated from the PCR products were analysed and assembled using Geneious Prime^®®^ 2023.0.1 (Biomatters Ltd., Auckland, New Zealand). Sequences were used for similarity BLAST searches in the National Centre for Biotechnology Information (NCBI) GenBank databases.

Thirty YMV coat protein (CP) sequences, representing YMV phylogenetic groups classified by Bousalem et al. [24] and Mendoza et al. [9], were downloaded from NCBI and aligned with 36 YMV CP sequences obtained from this study (Table 3) to generate a percentage similarity matrix using Multiple Alignment using Fast Fourier Transform (MAFFT) v7.490 in Geneious Prime^®®^ 2023.0.1. The aligned sequences were used for phylogenetic analysis using the Neighbor-Joining (NJ) method in Molecular Evolutionary Genetics Analysis across Computing Platforms (MEGA X) v10.2.6 software [30]. The reliability of the tree branches was evaluated by bootstrap test in 1000 replicates.

### 2.3. New LAMP Primer Design for YMV Detection

A multiple sequence alignment of 125 YMV CP sequences (downloaded from the NCBI GenBank database on 5 April 2021) was carried out using the Mafft Alignment v7.450 in Geneious Prime^®®^ 2021.1.1 (Biomatters Ltd., Auckland, New Zealand). A consensus sequence based on the alignment was used to design new LAMP primers using the Primer Explorer V5 software (http://primerexplorer.jp/e/) and visual adjustment of the primers’ position to avoid mismatches.

### 2.4. Detection of YMV by RT-LAMP

The same RNAs analysed by RT-PCR were used as templates in RT-LAMP. Two sets of primers were used (Table 2). Each RT-LAMP reaction was carried out in three replicates. The RT-LAMP assays were set up as 25 µL reactions containing 1X isothermal master mix (OptiGene, Horsham, UK), 0.2 µM forward and reverse outer primers (F3 and B3), 1.6 µM forward and reverse internal primers (FIP and BIP), 0.4 µM forward and reverse loop primers (LF and LB), and 2 µL of RNA template or crude extract. The assays were run in a Genie III LAMP machine (OptiGene) at 65 °C for 45 min. The subsequent melting process from 98 °C to 80 °C was carried out with a ramp rate of −0.05 °C/s.

### 2.5. Sensitivity Test for the Improved YMV RT-LAMP Assay

Purified total RNA (100 ng/µL) from a YMV-infected yam plant (Nig14) was serially diluted in RNA (100 ng/µL) from a YMV-negative plant (Nig15). Ten-fold serial dilutions down to 10^−9^ were tested by RT-LAMP and RT-PCR in duplicate assays. The sensitivity of the improved RT-LAMP for detecting YMV from crude RNA extracts was also evaluated. Similar to the purified total RNA, crude RNA extract from Nig14 was diluted ten-fold down to 10^−6^ with the crude extract from Nig15 and tested by RT-LAMP.

## 3. Results

### 3.1. Indexing of YMV by RT-PCR and RT-LAMP Assays

During routine testing of yam plants for YMV detection, discrepancies were found between the standard RT-PCR test and the RT-LAMP developed by Nkere et al. [23]. Three of six samples that tested positive by RT-PCR (Figure 1A), namely Gh3, Gh5, and Nig1, and that showed mild symptoms of YMV infection (Figure 2) tested negative by RT-LAMP (Figure 1B). The PCR products from Gh5 and Nig1 were sequenced (GenBank accession OQ677014 and OQ677015) and showed 99.1% and 98.9% identity, respectively, to *Yam mosaic virus* isolate DrCDI1, GenBank AJ305449. In addition to the negative RT-LAMP results with samples Gh3, Gh5, and Nig1, there were also late amplification times (>30 min) obtained for samples Gh1 and Gh2 (Figure 1B).

These unsatisfactory RT-LAMP results prompted the design of new LAMP primers to increase the specificity of the assay for YMV detection. New YMV LAMP primers (YMV1-OPT, Figure 3) were designed and used to test the same samples previously tested by RT-PCR and RT-LAMP using the Nkere et al. [23] primers. The new RT-LAMP using YMV1-OPT primers, subsequently referred to as the improved RT-LAMP test, detected YMV from all six samples within 15 min (Figure 4).

### 3.2. Evaluation of Improved YMV RT-LAMP Assay Specificity

RNA extracts from leaves of 14 *D. alata* plants were tested by RT-PCR for YMV and YMMV, another potyvirus infecting yam. Of these, 9/14 tested positive for YMMV only, 1/14 positive for YMV only, and 1/14 positive for both YMV and YMMV. PCR products of the YMV-positive samples were Sanger-sequenced by the Source BioScience sequencing service (Cambridge, UK), which confirmed the presence of YMV.

The same 14 *D. alata* RNAs were used to test the specificity of the improved RT-LAMP test. The assay detected YMV from the two YMV-positive samples, DA Nig1 and CTRT127, detected by RT-PCR (Table 4). All other samples were negative for YMV, confirming that there was no cross-reactivity of the YMV1-OPT primers with YMMV or the host plant.

### 3.3. Sensitivity of Improved YMV RT-LAMP

The sensitivity of the improved RT-LAMP assay for detecting YMV was compared to RT-PCR using primers by Mumford et al. [26]. RNA obtained from a YMV-infected *D. rotundata* plant was serially diluted ten-fold down to 10^−9^ using RNA from a YMV-negative *D. rotundata* plant. Each dilution was indexed for YMV by RT-PCR and the improved RT-LAMP assay. YMV positive amplifications were obtained from both assays down to 10^−9^ (Figure 5A,C). The time required to detect YMV in the most dilute sample (10^−9^) was approximately 32 min (Figure 5A). Similarly, serially diluted crude RNA extracts derived from incubating one YMV-infected leaf disc in PEG buffer were also tested for YMV via the improved RT-LAMP assay. YMV was detected in the sample RNAs diluted down to 10^−2^ (Figure 5B).

### 3.4. Comparison of Conventional RT-PCR and the New RT-LAMP

Purified total RNAs from 53 leaf samples of *D. rotundata* and *D. alata* were tested for YMV using the improved RT-LAMP assay and compared with conventional RT-PCR. A total of 36 samples tested positive for YMV by both tests (Table 5). With RT-LAMP, positive amplification signals were obtained in <26 min compared to >150 min required for RT-PCR. All samples that were negative by RT-LAMP were also negative by RT-PCR. The actin housekeeping gene was targeted by RT-PCR and used as an internal control to confirm the good quality of the RNAs, and YMV-negative results were due to a lack of viral RNA rather than any inhibition of the assay (results not shown).

### 3.5. Sequence Identity and Phylogenetic Analysis of YMV Amplicons

The mean pairwise nucleotide identity of Sanger-sequenced PCR products from YMV-positive samples (*n* = 36) obtained in this study was 97.1%. Nucleotide pairwise comparison of these sequences with YMV CP sequences downloaded from NCBI GenBank (*n* = 30) revealed 89.2–99% nucleotide identities, higher than the proposed International Committee on Taxonomy of Viruses (ICTV) criterion of <76–77% nucleotide identity for species demarcation of potyvirus CP gene [32,33].

Phylogenetic analysis clustered the YMV sequences from this study into six phylogenetic groups (Figure 6), with a percentage identity matrix of >97–100% within groups and <97% between groups (Appendix A). Isolates Ben1, Cam4, Gh2, Gh3, Gh15, Gh17, Gh18, Gh19, Gh20, Gh23, Gh27, Gh29, Gh30, Gh32, Gh33, Gh35, Gh36, Nig3, Nig4, Nig6, and Tog2, from samples collected from Benin, Cameroon, Ghana, Nigeria, and Togo, clustered in group III, an African group as classified by Bousalem et al. [24]. The other 16 isolates formed five new groups, XI, XII, XIII, XIV, and XV. Isolates Nig2, Tog1, Cam3, and Gh21 stood out as separate groups labelled as XI, XII, XIII, and XIV, respectively, and had pairwise nucleotide identities of <97% compared to sequences in other phylogenetic groups (Figure 6). Isolates Cam2, Gh5, Gh28, Gh34, Nig1, Nig5, Nig10, Nig11, Nig12, Nig13, and Nig14 clustered together with a YMV reference genome from Nigeria, MG711313 [8] to form group XV, which also had a pairwise nucleotide identity of <97% with sequences in other phylogenetic groups (Figure 6).

Sequences of YMV isolates obtained from this study were tested for recombination using the Recombination Detection Program (RDP) v.4.101. [34]. No recombination was detected among the YMV sequences.

## 4. Discussion

This study aimed to strengthen virus diagnostics in the seed yam systems by improving existing diagnostic tests because virus detection is crucial for efficient disease management in clean seed propagation systems, most notably during sanitation programs (review by Diouf et al. [3]). An RT-LAMP assay improved in both its specificity and sensitivity for YMV, one of the most economically damaging yam viruses globally, has in this study been developed to assist in the identifying of virus-free yam planting materials.

The false-negative results obtained by previously reported YMV LAMP primers [23] appear to be due to them having been designed from an alignment of the then available YMV coat protein sequences not fully encompassing diversity in the primer targeted regions. Aligning these primers to 125 YMV coat protein sequences revealed mismatches at the 3′ end of the primers (Appendix A). Studies have shown that 3′ terminal mismatches are detrimental to nucleic acid-based amplifications, resulting in a decreased amplification copy number or complete inhibition of amplification, hence providing false-negative results [35,36,37,38]. This prompted the need to develop a new YMV LAMP primer set.

The new RT-LAMP primer set, YMV1-OPT, demonstrated higher specificity than the existing RT-LAMP primer set [23], as it detected YMV from samples that tested negative with the existing primer set. This is assumed to be due to the YMV1-OPT primers having been designed to minimise the mismatches to 125 YMV sequences from the GenBank database and inserting degenerate codes at 3′ ends where mismatches could not be avoided [39,40,41].

Multiple primer combinations were evaluated during the design and selection process of the YMV1-OPT primers (results not shown). Mismatches were avoided at the 3′ ends to the greatest extent; however, where inevitable, nucleotide mismatches with <20% of the aligned sequences in one or two positions were tolerated for the outer and loop primers. Previous studies have shown that mismatches are better tolerated in the outer primers than inner primers [42,43]. This could be because the inner primers initiate the amplification process in LAMP assays, hence the need for specificity. The FIP and BIP primers are a fusion of F1C and F2, and B1C and B2 primers, respectively, with the F2 and B2 regions at the 3′ ends and F1C and B1C at the 5′ ends. Hence, mismatches at the 3′ ends of the F2 and B2 primers were replaced with degenerate codes, while those at the F1C and B1C regions were ignored.

Among the YMV isolates sequenced in this study, 21 clustered with isolates belonging to group III were reported in previous studies [9,24], while 11 clustered with a YMV reference genome, MG711313, from Nigeria [8]. According to the classification described by Bousalem et al. [24], Groups I, II, III, IV, VII, and IX comprise samples collected from the African region. An in-silico analysis of sequence alignments of the YMV1-OPT primers with representative sequences of these African groups indicates that the primers will detect YMV isolates from throughout the West African region. The remaining four YMV isolates, Gh21, Cam3, Tog1, and Nig2, formed four distinct groups, suggesting new phylogenetic groups, perhaps associated with isolates from other parts of the world. However, due to limited resources, only samples obtained from West Africa were used for this study. Further studies will be required to validate the detection of YMV from other yam-growing regions of the world using the YMV1-OPT primers.

The inability of RT-PCR to detect YMV from sample 4 (Figure 2) that tested positive by RT-LAMP could be due to PCR-inhibitory substances co-extracted with the yam RNA since the yam actin gene was also not detected in the same sample. Mumford and Seal [26] reported that yam tissues contain some PCR-inhibitory substances that could be co-extracted with the RNA. This suggests a higher tolerance of RT-LAMP to inhibitors than RT-PCR, which has also been reported in other studies [23,44,45].

Previous studies have shown that RT-LAMP is at least 100 times more sensitive than RT-PCR [23,46,47]. Nkere et al. [23] reported a sensitivity limit of 1000 fg/µL–100 fg/µL and 0.1 ng/µL–0.01 ng/µL for YMV LAMP primers, and YMV-F3x and YMV-B3x PCR primers, respectively. However, in this study, YMV1-OPT primers and YMV CP 1F and YMV UTR 1R PCR primers [25] both detected YMV from an infected sample down to the lowest dilution tested, 0.1 fg/µL (10^−9^). However, the differences in sensitivities could be due to variations in the YMV concentration of tested samples. YMV amplification was observed at ~32 min in the most dilute RNA sample, 0.1 fg/µL, suggesting that 40 min was sufficient to detect YMV in samples with low virus titre. The improved RT-LAMP assay also detected YMV from crude RNA extracts diluted down to 10^−2^, indicating that this approach can be used for rapid detection and in-field diagnosis of YMV. Overall, these imply that the YMV1-OPT primers are highly sensitive and valuable for laboratory-based and in-field detection of YMV.

Highly specific and sensitive diagnostic tests are required for the reliable diagnosis of plant viruses [22,26,28,48]. However, the development of diagnostic tests is an ongoing task. Diagnostic primers must be reviewed regularly and updated as new virus isolates are reported, as this would prevent false-negative results that might arise from potential diversity in such isolates [24,28,49], as observed in this study. The improved RT-LAMP assay will enhance the specificity of YMV detection in the production of virus-free seed yams in West Africa.

## 5. Conclusions

The routine detection of YMV via RT-LAMP using crude RNA extracts offers a significant cost and time-saving alternative to RT-PCR assays being used in the seed systems, which require extensive RNA extraction procedures. Furthermore, amplification products are visualised by monitoring the fluorescence generated by positive samples in real-time, thus reducing the likelihood of post-assay contaminations associated with PCR assays. This study presents an RT-LAMP assay with improved specificity and sensitivity for detecting YMV, which can be implemented at several stages of the seed multiplication process to eliminate YMV-positive samples quickly and cost-effectively. The YMV1-OPT primers designed in this study are being used to develop a ready-to-use YMV commercial kit (YMV1-OPT Isothermal kit, OptiGene). This will reduce the assay preparation time and the risk of contamination from the handling process.

## Figures and Tables

**Figure 1 viruses-15-01592-f001:**
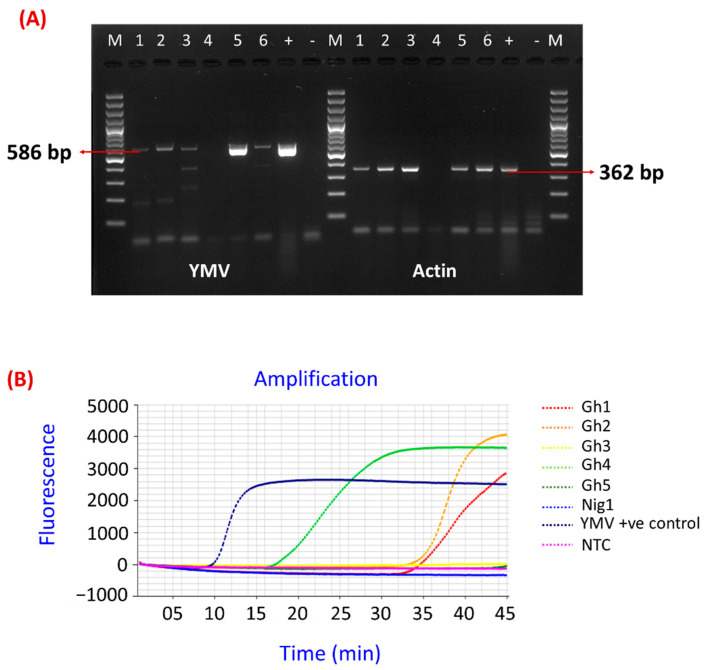
The detection of YMV in *Dioscorea rotundata* samples. (**A**)-The detection of YMV and actin by the reverse transcription polymerase chain reaction (RT-PCR) [22,25], M-100 bp Gene ruler DNA ladder size 100–3000 (ThermoFisher Scientific), Well 1-Gh1, 2-Gh2, 3-Gh3, 4-Gh4, 5-Gh5, 6-Nig1, + = YMV-positive control and NTC = non-template control; (**B**)-The detection of YMV by reverse transcription loop-mediated isothermal amplification (RT-LAMP) using primers by Nkere et al., 2018; NTC-Non-template control; +ve = YMV-positive control.

**Figure 2 viruses-15-01592-f002:**
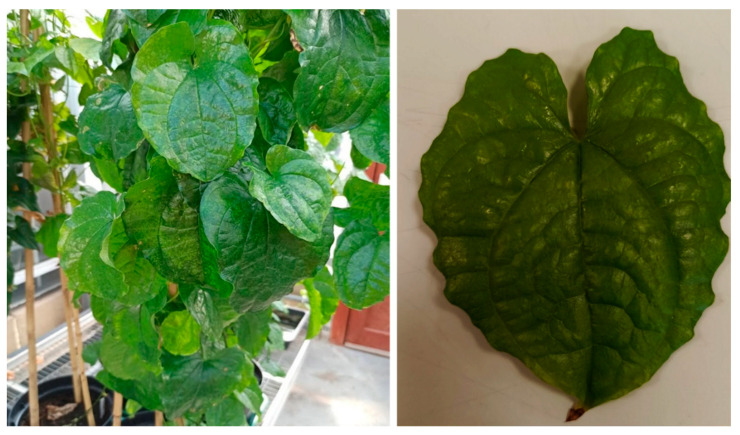
Yam plant (Nig1) showing mild mottle symptoms.

**Figure 3 viruses-15-01592-f003:**
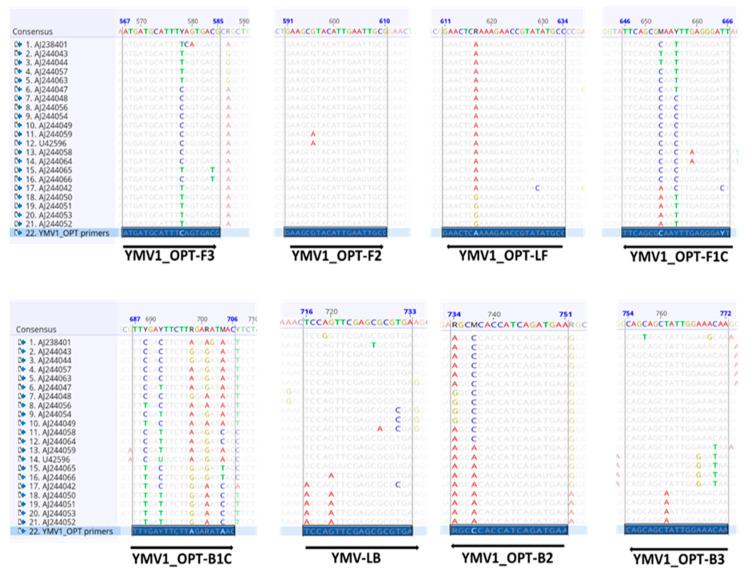
A representative figure showing a partial alignment of YMV coat protein sequences, highlighting the YMV1-OPT LAMP primers designed in this study. YMV1_OPT-F3 = Forward outer primer; YMV1_OPT-B3 = Reverse outer primer; YMV1_OPT-F1C + YMV1_OPT-F2 = Forward inner primer; YMV1_OPT-B1C + YMV1_OPT-B2 = Reverse inner primer; YMV1_OPT-LF = Forward loop primer; YMV1_OPT-LB = Reverse loop primer.

**Figure 4 viruses-15-01592-f004:**
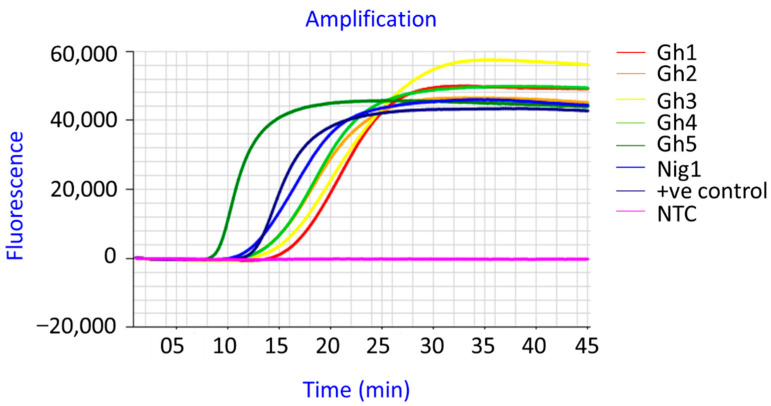
YMV detection by RT-LAMP using YMV1-OPT primers designed in this study. Template RNAs are the same tested by RT-PCR and RT-LAMP by Nkere et al. (2018); +ve control = YMV-positive control; NTC = Non-template control.

**Figure 5 viruses-15-01592-f005:**
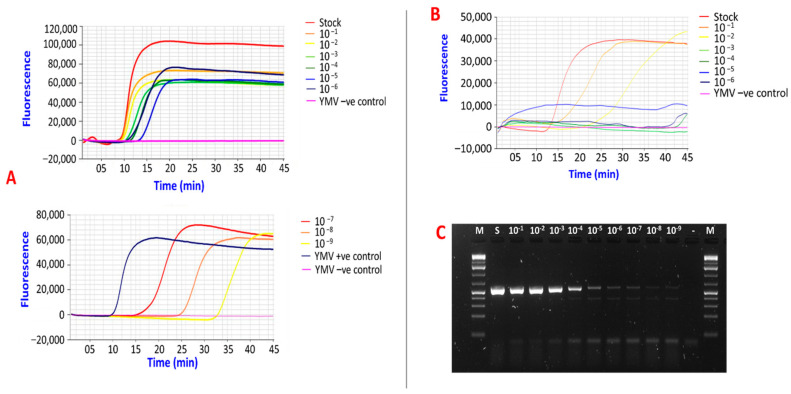
Sensitivity of the improved RT-LAMP and comparison with RT-PCR. (**A**)-YMV amplification from purified total RNA extracts by RT-LAMP using YMV1-OPT primers; Stock- Stock RNA from YMV-positive plant (100 ng/µL). The LAMP assay was conducted using the GenieIII machine (OptiGene), which allows only eight reactions in a run. (**B**)-YMV amplification from crude RNA extracts by RT-LAMP using YMV1-OPT primers. (**C**)-YMV amplification by RT-PCR from purified total RNA; M-1 kb DNA ladder, size 0.5 kb−10 kb (New England Biolabs); S-Stock RNA (100 ng/µL); - = YMV-negative control.

**Figure 6 viruses-15-01592-f006:**
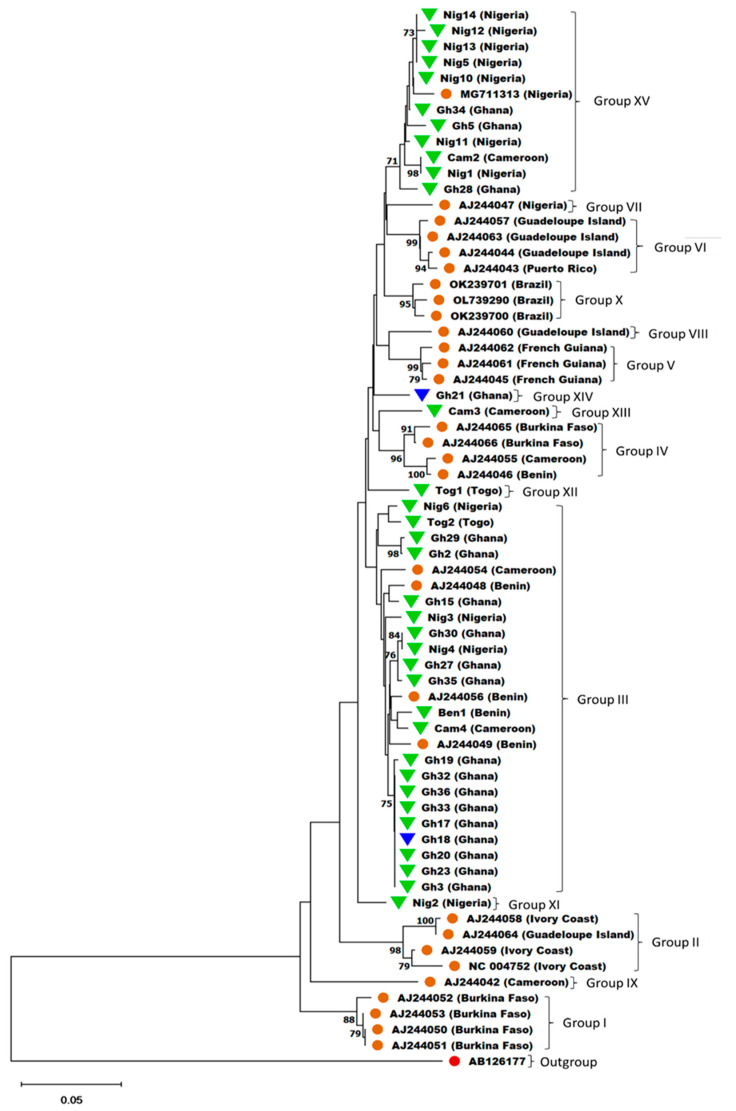
A phylogenetic tree based on the partial nucleotide sequence of the coat protein region of YMV. (▼) YMV CP sequences from *D. rotundata* obtained in this study; (▼) YMV CP sequences from *D. alata* obtained in this study; (**●**) YMV CP sequences from *D. rotundata* downloaded from NCBI; (**●**) the sequence of Pepper veinal mottle virus partial CP gene used as an outgroup. The tree was generated using the neighbour-joining method in MEGA-X with 1000 bootstrap replications. Branches < 70% were collapsed. The scale bar represents the number of nucleotide substitutions per site.

**Table 1 viruses-15-01592-t001:** Yam samples used in this study.

Sample ID	Collection Origin	*Dioscorea* spp.
Ben1	Benin	*D. rotundata*
Cam1	Cameroon	*D. rotundata*
Cam2	Cameroon	*D. rotundata*
Cam3	Cameroon	*D. rotundata*
Cam4	Cameroon	*D. rotundata*
Gh1	Ghana	*D. rotundata*
Gh2	Ghana	*D. rotundata*
Gh3	Ghana	*D. rotundata*
Gh4	Ghana	*D. rotundata*
Gh5	Ghana	*D. rotundata*
Gh6	Ghana	*D. rotundata*
Gh7	Ghana	*D. rotundata*
Gh8	Ghana	*D. rotundata*
Gh9	Ghana	*D. rotundata*
Gh10	Ghana	*D. rotundata*
Gh11	Ghana	*D. rotundata*
Gh12	Ghana	*D. rotundata*
Gh13	Ghana	*D. rotundata*
Gh14	Ghana	*D. rotundata*
Gh15	Ghana	*D. rotundata*
Gh16	Ghana	*D. rotundata*
Gh17	Ghana	*D. rotundata*
Gh18	Ghana	*D. alata*
Gh19	Ghana	*D. rotundata*
Gh20	Ghana	*D. rotundata*
Gh21	Ghana	*D. alata*
Gh22	Ghana	*D. alata*
Gh23	Ghana	*D. rotundata*
Gh24	Ghana	*D. rotundata*
Gh25	Ghana	*D. rotundata*
Gh26	Ghana	*D. rotundata*
Gh27	Ghana	*D. rotundata*
Gh28	Ghana	*D. rotundata*
Gh29	Ghana	*D. rotundata*
Gh30	Ghana	*D. rotundata*
Gh31	Ghana	*D. rotundata*
Gh32	Ghana	*D. rotundata*
Gh33	Ghana	*D. rotundata*
Gh34	Ghana	*D. rotundata*
Gh35	Ghana	*D. rotundata*
Gh36	Ghana	*D. rotundata*
Gh37	Ghana	*D. rotundata*
Nig1	Nigeria	*D. rotundata*
Nig2	Nigeria	*D. rotundata*
Nig3	Nigeria	*D. rotundata*
Nig4	Nigeria	*D. rotundata*
Nig5	Nigeria	*D. rotundata*
Nig6	Nigeria	*D. rotundata*
Nig7	Nigeria	*D. rotundata*
Nig8	Nigeria	*D. rotundata*
Nig9	Nigeria	*D. rotundata*
Nig10	Nigeria	*D. rotundata*
Nig11	Nigeria	*D. rotundata*
Nig12	Nigeria	*D. rotundata*
Nig13	Nigeria	*D. rotundata*
Nig14	Nigeria	*D. rotundata*
Nig15	Nigeria	*D. rotundata*
Tog1	Togo	*D. rotundata*
Tog2	Togo	*D. rotundata*

**Table 2 viruses-15-01592-t002:** YMV primers used in this study.

Test	Primer Name	Position *	Sequence (5’–3’)	Orientation ^#^	Reference
RT-LAMP	YMV1-OPT-F3	9120–9138	ATGATGCATTTCAGTGACG	F	This study
	YMV1-OPT-B3	9307–9305	TTGTTTCCAATAGCTGCTG	R	
	YMV1-OPT-FIP (F1C + F2)	F1C: 9199–9219	ARTCCCTCAARTTGCGCTGAA-	R	
		F2: 9144–9163	GAAGCGTACATTGAATTGCG	F	
	YMV1-OPT-BIP (B1C + B2)	B1C: 9240–9259	TTYGAYTTCTTAGARATAAC-	F	
		B2: 9287–9304	TTCATCTGATGGTGGGCY	R	
	YMV1-OPT-LF	9164–9187	GGCATATACGGTTCTTTTGAGTTC	R	
	YMV1-OPT-LB	9269–9286	TCCAGTTCGAGCGCGTGA	F	
RT-LAMP	F3	9038–9055	GACAATGATGGACGGTGC	F	[23]
	B3	9228–9248	GAAGTCAAACGCATATCTAGC	R	
	FIP (F1C + F2)	F1C: 9109–9134	ACTGAAATGCATCATTATCTGAC GAA-	R	
		F2: 9059–9076	GCAAGTGGAATACCCATT	F	
	BIP (B1C + B2)	B1C: 9144–9171	GAAGCATACATTGAATTGCGGAA CTCAA-	F	
		B2: 9206–9244	TGAGTAATCCCTCAAGTTG	R	
	LF	9079–9103	GGTTTGGCATTTTCTATGATCGGTT	R	
	LB	9186–9205	CCCCGATACGGTATTCAGCG	F	
RT-PCR	YMV-CP 1F	9026–9045	ATCCGGGATGTGGACAATGA	F	[26]
	YMV-UTR 1R	9590–9608	TGGTCCTCCGCCACATCAAA	R	

F3 and B3—Forward and reverse outer primers, respectively; FIP and BIP—Forward and reverse internal primer, respectively; LF and LB—Forward and reverse Loop primers, respectively. * Alignment position of primers with the reference YMV complete genome sequence (GenBank ref ID. NC_004752.1). ^#^ F and R—Forward and reverse orientation, respectively.

**Table 3 viruses-15-01592-t003:** YMV isolates used for phylogenetic studies.

Group*	Isolate	Sample Origin	*Dioscorea* spp.	Accession Number	Reference
I	BFC 56	Burkina Faso	*D. cayenensis-rotundata*	AJ244052	[24]
	C1/C3	Burkina Faso	*D. cayenensis-rotundata*	AJ244053	[24]
	BFC 51/C11	Burkina Faso	*D. cayenensis-rotundata*	AJ244050	[24]
	BFC 54	Burkina Faso	*D. cayenensis-rotundata*	AJ244051	[24]
II	CKA1/C11	Ivory Coast	*D. cayenensis-rotundata*	AJ244059	[24]
	CID3/C12	Ivory Coast	*D. cayenensis-rotundata*	AJ244058	[24]
	POGNON/C1	Guadeloupe island	*D. cayenensis-rotundata*	AJ244064	[24]
	U42596	Ivory Coast	*D. cayenensis-rotundata*	NC004752	[31]
III	CAM1/C1	Benin	*D. cayenensis-rotundata*	AJ244054	[24]
	B1/c1	Benin	*D. cayenensis-rotundata*	AJ244048	[24]
	CBE6b/C3	Benin	*D. cayenensis-rotundata*	AJ244056	[24]
	B14	Cameroon	*D. cayenensis-rotundata*	AJ244049	[24]
	Ben1	Benin	*D. rotundata*	OR004217	This study
	Cam4	Benin	*D. rotundata*	OR004218	This study
	Gh2	Ghana	*D. rotundata*	OQ677012	This study
	Gh3	Ghana	*D. rotundata*	OQ677013	This study
	Gh15	Ghana	*D. rotundata*	OQ677004	This study
	Gh17	Ghana	*D. rotundata*	OQ677006	This study
	Gh18	Ghana	*D. alata*	OQ677007	This study
	Gh19	Ghana	*D. rotundata*	OQ677008	This study
	Gh20	Ghana	*D. rotundata*	OQ677009	This study
	Gh23	Ghana	*D. rotundata*	OQ677011	This study
	Gh27	Ghana	*D. rotundata*	OR004219	This study
	Gh29	Ghana	*D. rotundata*	OR004229	This study
	Gh30	Ghana	*D. rotundata*	OR004220	This study
	Gh32	Ghana	*D. rotundata*	OR004223	This study
	Gh33	Ghana	*D. rotundata*	OR004225	This study
	Gh35	Ghana	*D. rotundata*	OR004222	This study
	Gh36	Ghana	*D. rotundata*	OR004224	This study
	Nig3	Nigeria	*D. rotundata*	OR004228	This study
	Nig4	Nigeria	*D. rotundata*	OR004221	This study
	Nig6	Nigeria	*D. rotundata*	OR004226	This study
	Tog2	Togo	*D. rotundata*	OR004227	This study
IV	SOA Ai/C1	Burkina Faso	*D. alata*	AJ244065	[24]
	SOA2/C2	Burkina Faso	*D. alata*	AJ244066	[24]
	CAM2/C31	Cameroon	*D. cayenensis-rotundata*	AJ244055	[24]
	174/C1	Benin	*D. cayenensis-rotundata*	AJ244046	[24]
V	G5/C10	French Guiana	*D. trifida*	AJ244062	[24]
	G13/C1	French Guiana	*D. trifida*	AJ244061	[24]
	GY/INRA/C11	French Guiana	*D. trifida*	AJ244045	[24]
VI	CGU1/C18	Guadeloupe island	*D. cayenensis-rotundata*	AJ244057	[24]
	GR/SAVANE/C4	Guadeloupe island	*D. cayenensis-rotundata*	AJ244063	[24]
VI	CGU2/C4	Guadeloupe island	*D. cayenensis-rotundata*	AJ244044	[24]
	AID 10/5	Puerto Rico	*D. alata*	AJ244043	[24]
VII	608	Nigeria	*D. cayenensis-rotundata*	AJ244047	[24]
VIII	DIVIN	Guadeloupe Island	*D. cayenensis-rotundata*	AJ244060	[24]
IX	CAM2	Cameroon	*D. cayenensis-rotundata*	AJ244042	[24]
X	YMV_DR2	Brazil	*D. cayenensis-rotundata*	OK239701	[9]
	YMV_DR1	Brazil	*D. cayenensis-rotundata*	OK239701	[9]
	YMV_I4	Brazil	*D. cayenensis-rotundata*	OL739290	[9]
XI	Nig2	Nigeria	*D. rotundata*	OR004232	This study
XII	Tog1	Togo	*D. rotundata*	OR004230	This study
XIII	Cam3	Cameroon	*D. rotundata*	OR004231	This study
XIV	Gh21	Ghana	*D. alata*	OQ677010	This study
	Cam2	Cameroon	*D. rotundata*	OR004209	This study
	Gh5	Ghana	*D. rotundata*	OQ677014	This study
	Gh28	Ghana	*D. rotundata*	OR004210	This study
	Gh34	Ghana	*D. rotundata*	OR004211	This study
	Nig1	Nigeria	*D. rotundata*	OQ677015	This study
	Nig5	Nigeria	*D. rotundata*	OR004213	This study
	Nig10	Nigeria	*D. rotundata*	OR004212	This study
	Nig11	Nigeria	*D. rotundata*	OR004215	This study
	Nig12	Nigeria	*D. rotundata*	OR004216	This study
	Nig13	Nigeria	*D. rotundata*	OR004214	This study
	Nig14	Nigeria	*D. rotundata*	OQ677016	This study
	YMV-NG	Nigeria	*D. rotundata*	MG711313	[9]

Group*-YMV phylogenetic group following classification by Bousalem et al. [24], Mendoza et al. [9] and this study.

**Table 4 viruses-15-01592-t004:** The detection of YMV from *Dioscorea alata* samples by RT-PCR and RT-LAMP assays.

Sample ID	RT-PCR		RT-LAMP
	YMMV	YMV	YMV
DA Nig1	+	+	+
DA Nig2	+	−	−
DA Nig3	−	−	−
DA Tog2	+	−	−
DA Tog3	+	−	−
VU709	+	−	−
VU711	+	−	−
VU715	−	−	−
VU717	+	−	−
VU724	+	−	−
VU740	−	−	−
VU746	+	−	−
CTRT127	−	+	+
CTRT268	+	−	−
YMV-positive control	−	+	+
Non-template control	−	−	−

**Table 5 viruses-15-01592-t005:** The detection of YMV in leaves of *Dioscorea rotundata* and *D. alata* via reverse transcription loop-mediated isothermal amplification (RT-LAMP) and reverse transcription polymerase chain reaction (RT-PCR).

S/N	Sample ID	*Dioscorea* spp.	RT-PCR	Improved RT-LAMP	
				YMV Status	Time (min:sec)
1	Gh6	*D. rotundata*	−	−	−
2	Gh7	*D. rotundata*	−	−	−
3	Gh8	*D. rotundata*	−	−	−
4	Gh9	*D. rotundata*	−	−	−
5	Gh10	*D. rotundata*	−	−	−
6	Gh11	*D. rotundata*	−	−	−
7	Gh12	*D. rotundata*	−	−	−
8	Gh13	*D. rotundata*	−	−	−
9	Gh14	*D. rotundata*	−	−	−
10	Gh15	*D. rotundata*	+	+	13:56
11	Gh16	*D. rotundata*	+	+	11:05
12	Gh17	*D. rotundata*	+	+	09:53
13	Gh18	*D. alata*	+	+	18:16
14	Gh19	*D. rotundata*	+	+	11:57
15	Gh20	*D. rotundata*	+	+	10:41
16	Gh21	*D. alata*	+	+	14:14
17	Gh22	*D. alata*	+	+	11:35
18	Gh23	*D. rotundata*	+	+	25:02
19	Gh24	*D. rotundata*	−	−	−
20	Gh25	*D. rotundata*	−	−	−
21	Gh26	*D. rotundata*	−	−	−
22	Gh27	*D. rotundata*	+	+	08:30
23	Gh28	*D. rotundata*	+	+	10:00
24	Gh29	*D. rotundata*	+	+	11:15
25	Gh30	*D. rotundata*	+	+	09:15
26	Gh31	*D. rotundata*	+	+	20:00
27	Gh32	*D. rotundata*	+	+	08:30
28	Gh33	*D. rotundata*	+	+	09:00
29	Gh34	*D. rotundata*	+	+	10:15
30	Gh35	*D. rotundata*	+	+	09:30
31	Gh36	*D. rotundata*	+	+	10:00
32	Gh37	*D. rotundata*	−	−	−
33	Nig2	*D. rotundata*	+	+	08:15
34	Nig3	*D. rotundata*	+	+	10:00
35	Nig4	*D. rotundata*	+	+	25:45
36	Nig5	*D. rotundata*	+	+	08:00
37	Nig6	*D. rotundata*	+	+	10:45
38	Nig7	*D. rotundata*	−	−	−
39	Nig8	*D. rotundata*	−	−	−
40	Nig9	*D. rotundata*	−	−	−
41	Nig10	*D. rotundata*	+	+	10:00
42	Nig11	*D. rotundata*	+	+	13:00
43	Nig12	*D. rotundata*	+	+	09:45
44	Nig13	*D. rotundata*	+	+	09:30
45	Nig14	*D. rotundata*	+	+	10:05
46	Nig15	*D. rotundata*	−	−	−
47	Ben1	*D. rotundata*	+	+	08:00
48	Tog1	*D. rotundata*	+	+	07:15
49	Tog2	*D. rotundata*	+	+	10:45
50	Cam1	*D. rotundata*	+	+	08:00
51	Cam2	*D. rotundata*	+	+	07:30
52	Cam3	*D. rotundata*	+	+	12:30
53	Cam4	*D. rotundata*	+	+	09:45

## Data Availability

All sequences have been deposited in GenBank, accession numbers can be found in Table 3.

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
