# Peer review of "Improved Reverse Transcription Loop-Mediated Isothermal Amplification (RT-LAMP) for the Rapid and Sensitive Detection of *Yam mosaic virus"

_viruses, 2023, doi:10.3390/v15071592_

Round 1
Reviewer 1 Report
This article: Improved reverse transcription–loop-mediated isothermal amplification (RT-LAMP) for rapid and sensitive detection of Yam mosaic virus in seed yam systems, presents novel information and deserve to be considered for publication. I have few points to make which may be addressed while preparing the revision. RT-LAMP does not require thermocycling and is commonly used to detect RNA viruses, which can be visualization in outside of the laboratory with simple equipment. But there was not any visualization experiment from this paper.
Author Response
Thank you for your taking the time to review our manuscript. This is a good point. However, in our work, we used a GenieIII (Optigene) machine to develop the improved LAMP assay as it offers a higher level of precision and accuracy compared to colour change visualisation, which can be influenced by subjective interpretations. By employing the GenieIII machine, we can ensure reliable and objective fluorescence measurements throughout the study. Furthermore, the GenieIII is portable and built with an internal rechargeable battery suitable for field testing and adding further protection from equipment failure due to irregular power supplies commonly experienced in many yam-growing regions of the world, including in West Africa.
Reviewer 2 Report
The article is good but the title is misleading. If you read the title it seems that it is going to detect the virus in the seeds and it is not like that, it is done all the time in leaves. Therefore, I think it would be convenient to change the title.
Author Response
We appreciate your valuable feedback.
We understand that the title might create confusion, and we have revised it accordingly. The new title of the manuscript is “Improved reverse transcription–loop-mediated isothermal amplification (RT-LAMP) for rapid and sensitive detection of Yam mosaic virus (YMV)”.